



# The role of aerosol-cloud interactions in linking anthropogenic pollution over southern West Africa and dust emission over the Sahara

Laurent MENUT[1], Paolo TUCCELLA[2], Cyrille FLAMANT[3], Adrien DEROUBAIX[1,3], and Marco GAETANI[3,4]

[1]Laboratoire de Météorologie Dynamique, Ecole Polytechnique, IPSL Research University, Ecole Normale Supérieure, Université Paris-Saclay, Sorbonne Universités, CNRS, Route de Saclay, 91128 Palaiseau, France
[2]CETEMPS, Departement of Physical and Chemical Sciences and Center of Excellence in Telesening of Environment and Model Prediction of Severe Events, University of L'Aquila, Italy.
[3]LATMOS/IPSL, Sorbonne Université, UVSQ, CNRS, 75252 Paris, France
[4]Scuola Universitaria Superiore IUSS, Pavia, Italy

**Correspondence:** Laurent Menut, menut@lmd.polytechnique.fr

**Abstract.** The aerosol direct and indirect effects are studied over West Africa in the summer of 2016 using the coupled WRF-CHIMERE regional model including aerosol-cloud interaction parametrization. First, a reference simulation is performed and compared with observations acquired during the Dynamics-Aerosol-Chemistry-Cloud Interactions in West Africa (DAC-CIWA) field campaign which took place in June and July 2016. Sensitivity experiments are also designed to gain insights
into the impact of the aerosols dominating the atmospheric composition in southern West Africa (one simulation with halved anthropogenic emissions and one with halved mineral dust emissions). The most important effect of aerosol-cloud interactions is found for the mineral dust scenario and it is shown that halving the emissions of mineral dust decreases the 2-m temperature by 0.5 K and the boundary layer height by 25 m in monthly average and over the Saharan region. The presence of dust aerosols also increases (resp. decreases) the shortwave (resp. longwave) radiation at the surface by 25 W/m$^2$. It is also shown that the
decrease of anthropogenic emissions along the coast has an impact on the mineral dust load over West Africa by increasing their emissions in Saharan region. It is due to a mechanism where particulate matter concentrations are decreased along the coast, imposing a latitudinal shift of the monsoonal precipitation, and, in turn, an increase of the surface wind speed over arid areas, inducing more mineral dust emissions.

## 1 Introduction

Megacities in the Gulf of Guinea are under frequent and intense air pollution episodes, with pollutants mostly originating from local anthropogenic emissions as well as form a variety of remote sources such as the Sahara and the Sahel (mineral dust), and Central Africa (biomass burning products). This atmospheric pollution has an impact on human health, (Bauer et al., 2019), and climate, but also, in the short term, on meteorology and radiation through the direct and indirect effects of aerosols, (Haywood and Boucher, 2000; Andreae and Rosenfeld, 2008). These interactions are complex, not completely known and many studies



are currently investigating this relationship to quantify how aerosols can affect meteorology and radiation. These studies cover many different scientific questions from hourly air quality (Yu et al., 2014; Forkel et al., 2015; Zhao et al., 2017) to long-term climate impacts (Mahowald et al., 2003; Luo et al., 2003).

Emitted mainly from Sahara and Sahel, mineral dust is of great interest for the aerosol-radiation-cloud interactions (ARCI). Its abundance as well as its absorbing properties in the shortwave and longwave affect massively the radiation in the atmo-
spheric column. Helmert et al. (2007) quantified the direct and semi-direct effect of Saharan dust over northern Africa and Europe and showed a decrease of 2-m temperature. After correction of the dust absorption used in models, Balkanski et al. (2007) showed, with the LMDz-INCA global model, that the dust radiative effect strongly depends on the brightness of the surface: over oceans and deciduous surfaces (albedo <0.15), dust will cool the atmospheric column. Over desert (albedo >0.3), dust will warm the atmospheric column. Using measurements, di Sarra et al. (2011) showed a large dust effect on shortwave
and longwave radiation when dust plumes overpass the Lampedusa Island. Rémy et al. (2015) quantified the feedbacks between free-troposphere dust layers and boundary-layer meteorology and showed that maximum temperatures are reduced, increasing atmospheric stability, then decreasing 10-m wind speed during daytime. The increase in atmospheric stability is also studied by Guo and Yin (2015), showing a decrease on East Asia precipitation as well as a reduction of the monsoon intensity due to a decrease of the land-sea temperature gradient. Mineral dust impacts precipitations because it is a large contributor to Cloud
Condensation Nuclei (CCN) and Ice Nuclei (IN) formation, (DeMott et al., 2010; Andreae and Rosenfeld, 2008). Recent studies show a variable impact on clouds and precipitation depending on the dust composition, size of the particles or altitude of the plume. In the upper troposphere, Hande et al. (2015),Nickovic et al. (2016) and Weger et al. (2018) showed the importance of mineral dust to create ice clouds.

Biomass burning aerosols also have a large impact on ARCI. Over a large model domain encompassing Central Africa (where
large biomass burning sources are observed in the boreal summer), the Gulf of Guinea and the Saharan region, Menut et al. (2018) showed that biomass burning aerosols may be transported towards West Africa following two different pathways (over the South-East Atlantic and over the continent), but, that ultimately, they would always impact the air quality of megacities such as Abidjan and Lagos. For large fires in Canada, Walter et al. (2016) showed that below the fire plume, the 2-m temperature may be decreased by 6 K due to the direct effect. During the 2010 Russian heat waves, cause of large fires, Baro et al. (2017)
showed that lofted biomass burning aerosols can reduce by 10% the 10-m wind speed. Hodzic and Duvel (2018) studied the impact of biomass burning aerosols on meteorology in Borneo and showed that aerosols tend, in average, to suppress warm rain and reduce the deep convection. This is achieved with different sensitivity tests on the biomass burning aerosol absorption, showing that precipitation is very sensitive to this parameter. Gordon et al. (2018) modelled the impact of biomass burning on the dynamics close to the Ascension Island, in the south-east Atlantic and showed that fires reduce the inversion height over
the ocean and strongly impact the radiative effect. But they note this impact is also sensitive to the model resolution as well as the way the rain autoconversion is taken into account.

During the West African Monsoon (WAM), (Parker et al., 2005), the ARCI are more complex than during the boreal winter and are not currently well known. This is partly due to the installation of deep convection inland leading to the development of mesoscale convective systems (MCS) and frequent precipitation events. It was studied during the AMMA project (Redelsperger



et al., 2006) and was at the heart of the recent *"Dynamics-aerosol-chemistry-cloud interactions in West Africa"* (DACCIWA) project, (Knippertz et al., 2015a, b, 2017; Flamant et al., 2018). Zhao et al. (2011) showed that mineral dust has a cooling effect at the surface and a warming effect in the troposphere. This leads to an increase of atmospheric stability during the day but a decrease during the night. It has an impact (but moderate) on precipitation by reducing the late afternoon precipitation but increasing those of the morning. Shi et al. (2014) studied the impact of a MCS and quantified the decrease in precipitation

due to the indirect effect and a delay in the precipitation event due to the direct effect. The large scale effect of aerosol on precipitation of the WAM was studied by Huang et al. (2009) where they showed a reduction of precipitation of 1.5 mm day$^{-1}$ (at the maximum). Using MODIS and CALIPSO satellite measurements, (Costantino and Bréon, 2013) also studied the aerosol indirect on warm clouds over the south-east Atlantic and showed a decrease of the cloud droplet radius, but also in cloud liquid water path probably due to a dry air entrainment at cloud top.

In the framework of the DACCIWA project, the impact of long-range transport of dust and biomass burning on surface pollution (gas and aerosols) was quantified using measurements and the WRF/CHIMERE models in Menut et al. (2018). However, this was done without taking into account the interactions between aerosols and clouds and for the summer of 2014, so that a comparison with extensive measurements was not possible. Deetz et al. (2018a) used the COSMO-ART online coupled model and analyzed the radiative impact of aerosols on liquid water content during the month of July 2016 and over the Gulf of

Guinea. They quantified the important impact of aerosol on shortwave radiation (a decrease of -20 W m$^{-2}$) whereas the impact on longwave radiation was found to be negligible. In the continuity of this work, Deetz et al. (2018b) showed that the amount of aerosols over southern West Africa impacts the dynamics of the daytime coastal moist front generated along the Gulf of Guinea and the intensity of its inland propagation, this feature modulating the transport of anthropogenic aerosols emitted at the coast (Deroubaix et al., 2019).

In the present study, we use the coupled regional model WRF-CHIMERE, including an aerosol-cloud interaction parametrization, to quantify the ARCI over the Gulf of Guinea within a modelled domain much larger than in Menut et al. (2018), in order to take into account the spatial and temporal variability of remote aerosol sources such as the Sahara or Central Africa. In section 2, we present the observations used to estimate the realism of our simulations. In section 3, we present the details of the online coupled model used as well as the different ways designed to quantify the ARCI. In section 4, the reference

simulation is compared to observations. In section 5, we perform simulations using emissions scenarios to estimate the ARCI when individual sources (dust and anthropogenic aerosols) are modified in a realistic manner. Conclusions are presented in section 7.

## 2   The measurements data

A large part of this study is focused on model-model comparisons. However, in order to evaluate the realism of the reference

simulation, we compare the modelled outputs to *AErosol RObotic NETwork* (AERONET) and *Met Office Integrated Data Archive System* (MIDAS) measurements. The data of these stations were previously used in Menut et al. (2018) for the study





of biomass burning plumes transport in the Gulf of Guinea and it has been shown that they are well adapted for this kind of studies with large modelled domains.

Aerosol Optical Depth (AOD) level 2 measurements are used from the AERONET dataset, (Holben et al., 2001). Comparisons between measurements and model outputs are performed using hourly time series of AOD at a wavelength of $\lambda$=550 nm.

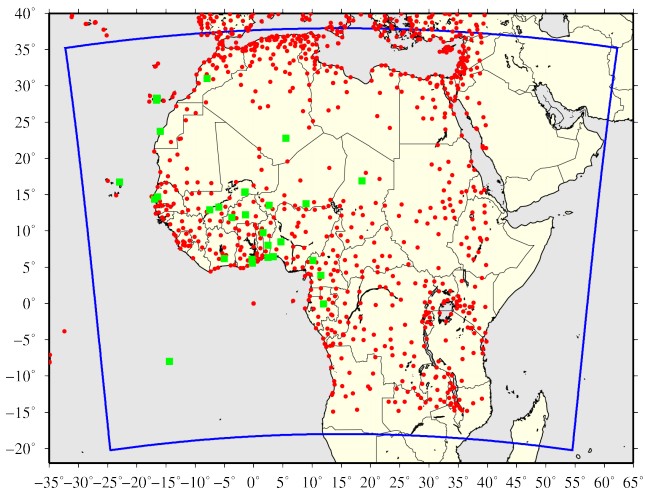

**Figure 1.** *Map of the studied domain. The blue frame represents the modelled domain. The red dots correspond to the Met Office MIDAS meteorological surface stations. The green dots correspond to the AERONET stations.*

The MIDAS meteorological surface stations data, (MetOffice, 2012), are used for precipitation rate, 2-m temperature and 10-m wind speed measurements. These observations are available in a three-hourly format and for many locations. For 10-m wind speed and 2-m temperature, instantaneous values are provided, while for precipitation rates, cumulated data over varying durations are proposed. All stations are displayed in Figure 1 even if data are not always available for all stations and all hours.

## 3 The modelling system

### 3.1 The WRF and CHIMERE models

To quantify interactions between aerosols, clouds and radiation, numerous online models were recently developed. These models are extensively presented and discussed in the reference papers of Zhang (2008) and Baklanov et al. (2014). In this study, we use a new *online access model* built with the Weather and Research Forecasting (WRF), (Powers et al., 2017), and CHIMERE, (Mailler et al., 2017), regional models. This coupling was developed following two phases: the direct effect by (Briant et al., 2017) and the indirect effect by (Tuccella et al., 2019). The choice for the coupling approach was to be the least intrusive possible in the two models and to implement a dialog between them by using an external coupler, the OASIS-MCT





tool, (Craig et al., 2017). For the direct effects, CHIMERE send to WRF the Aerosol Optical Depth (AOD), Single-Scattering

Albedo (calculated with the fastJX online model), Asymmetry factor. For the indirect effects, CHIMERE send to WRF the aerosol number size distribution, the hygroscopic aerosol number size distribution, the aerosols bulk hygroscopicity and the ice nuclei. The variables are exchanged with a frequency of 30 min.

The WRF regional model is used in its version 3.7.1 and calculates the meteorological variables. The model configuration is the same as in Menut et al. (2018). The global meteorological analyses from the National Centers for Environmental Pre-

diction (NCEP) with the Global Forecast System (GFS) products are used to nudge regional fields for pressure, temperature, humidity and wind. The spectral nudging approach is used, (Von Storch et al., 2000), for wavelengths greater than ≈2000km, corresponding to wave numbers less than 3 in latitude and longitude, for wind, temperature and humidity and only above 850 hPa. This configuration allows the regional model to create its own thermodynamics within the boundary layer. The large scale follows the thermodynamics fields from the NCEP analyses.

The CHIMERE chemistry-transport model calculates the concentrations of the gaseous and aerosols species. WRF and CHIMERE use the same horizontal grid with a 60 km × 60 km resolution to avoid interpolation during the coupling. The output results are issued hourly. The modelled period ranges from 15 June to 31 July 2016. We consider the first two weeks as spin-up time and the results are analyzed from 1 to 31 July 2016.

## 3.2 The aerosol, radiation, clouds interactions

The direct effect considers the scattering and absorption of solar and telluric radiation by aerosols, (Haywood and Boucher, 2000; Helmert et al., 2007; Zhang, 2008). It impacts the atmospheric dynamics below an aerosol layer by modifying the temperature and wind speed. In our model configuration, the direct effect is taken into account by estimating the AOD, the single scattering albedo (SSA) and the Asymmetry factor with CHIMERE and sending them to WRF and the Rapid Radiative Transfer Model for General Circulation Models (RRTMG) radiative transfer scheme (Iacono et al., 2008). Some direct effects were

already quantified with WRF/CHIMERE in Briant et al. (2017), studying the interactions between mineral dust concentrations and temperature in Africa.

The indirect effect takes into account the aerosol-induced increase in CCN and IN as well as the subsequent changes in clouds properties. An increase of CCN induces an enhanced cloud albedo, a longer lifetime of clouds (first indirect effect) and an enhanced cloud reflectivity due to suppression of precipitation (second indirect effect), (Andreae and Rosenfeld, 2008). The

grid-resolved cloud microphysics parameterization used in WRF is the aerosol-aware scheme of Thompson and Eidhammer (2014). This scheme calculates the cloud droplet nucleation rate using the aerosol size distribution calculated in CHIMERE. CHIMERE using a sectional approach for aerosol, the activation scheme of Thompson and Eidhammer (2014) is replaced by the one of Abdul-Razzak and Ghan (2002) in the present version of the coupled model. For the cloud ice formation, it is calculated in the Thompson and Eidhammer (2014) using the IN calculated in CHIMERE. For IN estimation, only mineral

dust concentrations with mean mass median diameter $D_p > 0.5 \mu m$ are taken into account. The scheme for heterogeneous ice nucleation is the one of DeMott et al. (2015). Homogeneous freezing of deliquesced aerosols is parameterized as in Thompson and Eidhammer (2014) following the method of (Koop et al., 2001). The climatology of deliquesced aerosol number concen-





tration is replaced with the CHIMERE prediction and is based on a mixture of hygroscopic particles with the diameter larger than 0.1 $\mu$m. Further details about the implementation of the aerosol indirect effects within WRF-CHIMERE are provided by

Tuccella et al. (2019).

Finally, it is important to notice that there are several limitations in the way the coupling is modeled, due to several scientific and technical locks. These limitations mainly concern the indirect effects and will all lead to an underestimation of the indirect effects in the simulations presented in the following. Aerosol indirect effects are implemented only in the grid-scale resolved clouds via the Thompson and Eidhammer (2014) scheme. Convective (subgrid) clouds are not affected by aerosol effects. Over

areas such as the Gulf of Guinea, clouds are mainly generated by convection, thus calculated by convection parameterizations. In our model configuration, convective clouds are treated by using the aerosol aware parameterization of Grell and Freitas (2014), but the indirect effect is not yet implemented in this scheme (as in many regional models). Some other schemes were also developed to take into account indirect effect in cumulus parameterizations such as Lim et al. (2013) and Berg et al. (2015). One can expect to underestimate the indirect effect due the missing part of coupling in the convective scheme. Another point

is the use of the nudging: the regional model is nudged in the global model, this one not considering interactive aerosol effects, (He et al., 2017).

### 3.3 Definition of the simulations

There are several ways to quantify the effect of aerosols on meteorology. It depends on the definition of the simulation dedicated to compare the results to the reference simulation (which uses aerosol-radiation-cloud interactions). In this study, we will

focus on a 'scenario' approach. This methodology sends modified aerosol concentrations to the meteorological model. This approach was used, for example, by Lim et al. (2013) to investigate the indirect effect. With this approach, results have the order of magnitude of realistic changes in the atmosphere. Differences are calculated between a 'reference' simulation and scenario simulations. The 'reference' simulation is called CPLfull and contains the full emissions and the ARCI. The two scenario simulations are "CPLanthro" and "CPLdust" with the ARCI and halved anthropogenic and mineral dust emissions,

respectively.

### 4  Comparison to observations

The first step is to compare the CPLfull simulation results to available MIDAS, AERONET and soundings observations.

### 4.1  Definition of statistical scores

Three statistical indicators are used: the spatial Pearsons' correlation, the temporal Pearsons' correlation and the normalized

RMSE. The temporal correlation, $R_t$, is computed for each station and is directly related to the hourly variability. $O_{t,i}$ and $M_{t,i}$





represent the observed and modelled values, respectively, at time $t$ and for the station $i$, for a total of $T$ days and a total of $I$ stations. The mean time averaged value $\overline{X_i}$ is:

$$\overline{X_i} = \frac{1}{T}\sum_{t=1}^{T} X_{t,i} \tag{1}$$

The temporal correlation $R_{t,i}$ for each station $i$ is calculated as:

$$R_{t,i} = \frac{\sum_{t=1}^{T}(M_{t,i} - \overline{M_i})(O_{t,i} - \overline{O_i})}{\sqrt{\sum_{t=1}^{T}(M_{t,i} - \overline{M_i})^2 \sum_{t=1}^{T}(O_{t,i} - \overline{O_i})^2}} \tag{2}$$

The mean temporal correlation, $R_t$, used in this study is thus:

$$R_t = \frac{1}{I}\sum_{i=1}^{I} R_{t,i} \tag{3}$$

The spatial correlation, noted $R_s$, uses the same formula type except it is calculated from the temporal mean averaged values of observations and model for each location where observations are available.

The spatio-temporal mean averaged value is calculated as:

$$\overline{\overline{X}} = \frac{1}{I}\sum_{i=1}^{I} \overline{X_i} \tag{4}$$

and the spatial correlation, $R_s$:

$$R_s = \frac{\sum_{i=1}^{I}(\overline{M_i} - \overline{\overline{M}})(\overline{O_i} - \overline{\overline{O}})}{\sqrt{\sum_{i=1}^{I}(\overline{M_i} - \overline{\overline{M}})^2 \sum_{i=1}^{I}(\overline{O_i} - \overline{\overline{O}})^2}} \tag{5}$$

The normalized Root Mean Square Error, $nRMSE$, is expressed as:

$$nRMSE = \sqrt{\frac{1}{T}\frac{1}{I}\sum_{t=1}^{T}\sum_{i=1}^{I}\left(\frac{O_{t,i} - M_{t,i}}{O_{t,i}}\right)^2} \tag{6}$$

for all stations $i$ and all times $t$.

## 4.2 MIDAS

The comparison between model and observations is made on a daily basis. As data from some stations are not always continuously available, the comparison is carried out on a day by day basis, keeping in mind that some days may average a more

complete diurnal cycle than others.





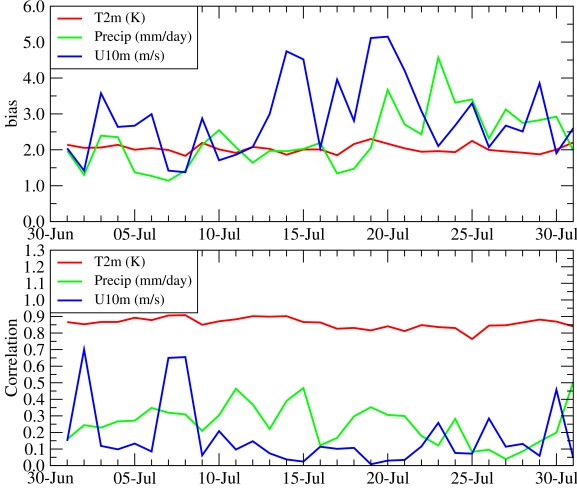

**Figure 2.** *Time series of bias 'CPLfull minus observations' and spatial correlation between CPLfull and observations) for 2-m temperature (K), 10-m wind speed (m s⁻¹) and total (convective+stratiform) precipitation rate (mm/day) for the month of July 2016.*

Results of statistical scores are presented in Figure 2 as time series of daily-averaged variables. For the 2-m temperature, the bias is around $2\,\mathrm{K}$ and fairly constant. The spatial correlation has high values (between 0.75 and 0.9) showing that this variable is well simulated (also knowing that correlation is positively influenced by the latitudinal effect). For the 10-m wind speed, statistical scores are less good than for 2-m temperature. The bias is important, between 1 and $5\,\mathrm{m\,s^{-1}}$, and varies a lot from one day to another. The same behaviour is noted for the spatial correlation, with low values of correlation (mostly around 0.1) and a high day-to-day variability. For the total precipitation, there is no general tendency, the bias being between 1 and 5 $\mathrm{mm\,day^{-1}}$.

Compared to the state of the art of simulations over this domain, there is no large statistical improvement when adding the effect of aerosols on wind and temperature. It is not completely surprising, the region being under a meteorology where convection and precipitation are important. Thus, the impact of aerosols is not the predominant process in the atmosphere during the monsoon period. The same kind of conclusion was presented by Baró et al. (2017) for analysis and comparison to E-OBS data over Europe: they showed the bias and RMSE scores for 2-m temperature were improved when ARCI is taken into account (using a large set of regional online coupled models) but not enough to explain the gap between model and observations.

## 4.3  AERONET

The AOD time series are presented in Figure 3 for some selected stations where hourly model results are compared to AERONET data: Cinzana, Saada and Savè. These stations are selected for their location representative of different environments (for Cinzana and Saada) and for Savè because it was a super-site during the DACCIWA campaign. The order of





magnitude is correctly reproduced except in Savè where a significant low bias appears. For all sites, all peaks are well mod-
elled, showing that transport of aerosols is correctly performed. The Angström exponent (ANG) is well modelled, showing that
the nature of the aerosols is also well estimated.

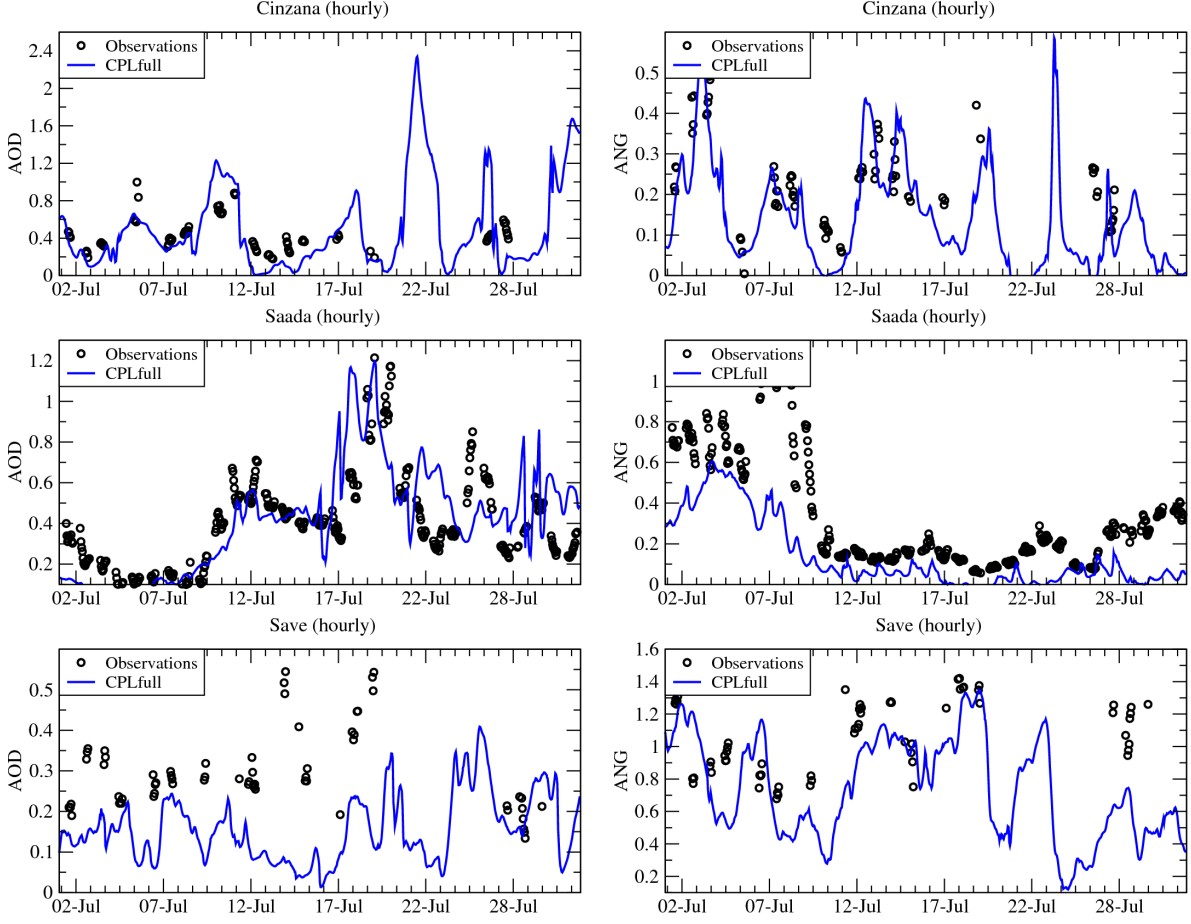

**Figure 3.** *Aerosol Optical Depth (at wavelength λ=600nm, left column) and Angström exponent (right column) time series in Cinzana, Saada
and Savé and for July 2016.*

## 4.4 Soundings

Comparisons between observation and model output in the first 4 km of the troposphere are presented in Figure 4. Observations
are in the left panel and represent soundings made in several places: Abidjan, Accra, Cotonou, Savè, Lamto and Parakou. For
each location, all soundings recorded between 1 and 31 July are averaged. The same is done for the model outputs, where
hourly gridded data are interpolated to fit the time and location of the observations (central panel). This methodology was




already used in (Deroubaix et al., 2019) for similar vertical profiles. All information about the soundings and the experimental campaign are in (Flamant et al., 2018). Finally, the difference (mod-obs) is displayed in the right panel.

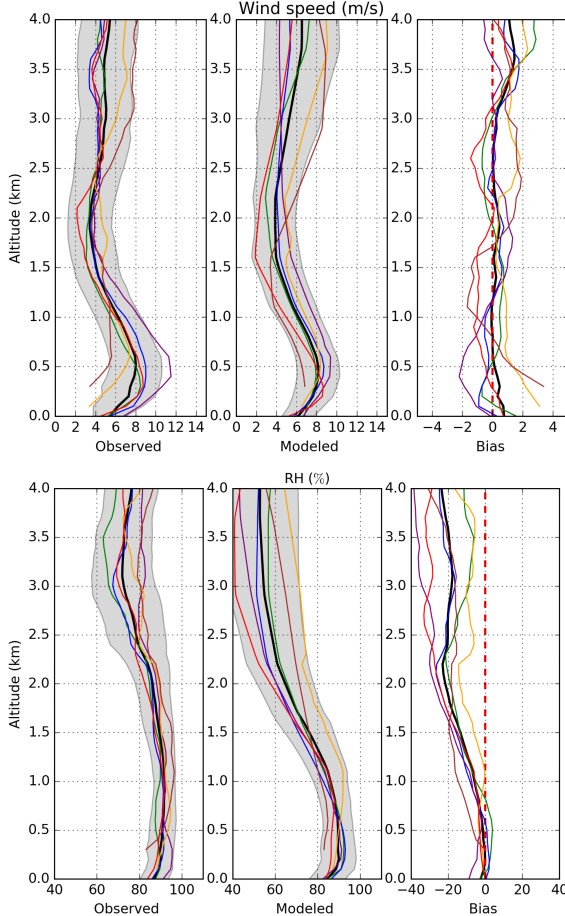

**Figure 4.** *Observed and modeled mean vertical profiles of wind speed (in m s$^{-1}$) and relative humidity (RH in %) averaged of all profiles over the period 1-31 July 2016 at Abidjan in Ivory Coast (green line), Accra in Ghana (blue line), Cotonou in Benin (purple line), Savè in Benin (orange line), Lamto in Ivory Coast (red line) and Parakou (Benin). The mean and standard deviation at the four locations are represented by the black line and the gray shading, respectively. Outputs from the WRF model are interpolated along the radiosonde positions. The right panel presents the (mod-obs) mean vertical bias at each location and of the average of the four locations. The dashed line materializes the boundary between negative and positive bias.*

For the wind speed, and at all sites, it is shown that a local maximum is present at 500 m above ground level (AGL) with values between 8 and 12 m s$^{-1}$. These profiles present the same structure as described in (Deroubaix et al., 2019) with three clearly defined vertical layers from the surface to 4 km AGL: (i) the monsoon layer in the first kilometer (with a maximum at 0.5 km AGL); (ii) a vertical wind shear layer from 1 to 2 km AGL in which wind speed decreases with altitude to a minimum





around 2 km AGL and (iii) a layer above in which wind speed is increasing with altitude and is influenced by the presence of the African Easterly Jet (AEJ). The overall structure of the wind speed profile is well reproduced in the simulation. The bias is

comprised between -2 and 2 $\mathrm{m\,s^{-1}}$ and highly variable in terms of location and height.

The relative humidity is well reproduced in the monsoon layer but there is a dry systematic bias from 1 to 4 km in the simulation, which leads to a probable under-estimation of liquid water content, thus the cloud cover and the indirect effects.

### 4.5 Synthesis of scores

The statistical results are calculated with hourly data over the whole month of July 2016. They are presented in Table 1 for 2-m

temperature, 10-m wind speed, precipitation rate, AOD and ANG.

| Variable | $R_s$ | $R_t$ | nRMSE | bias |
|---|---|---|---|---|
| 2m temperature (K) | 0.87 | 0.68 | 0.12 | -0.29 |
| 10-m wind speed (m/s) | 0.79 | 0.25 | 1.20 | 1.26 |
| Precipitation (mm/day) | 0.64 | -0.09 | 3.18 | -0.21 |
| AOD | 0.44 | 0.40 | 2.64 | 0.14 |
| ANG | 0.97 | 0.50 | 0.76 | -0.17 |

**Table 1.** *Statistical scores (correlation, RMSE and bias) for the 2-m temperature, 10-m wind speed, precipitation rate, Aerosol Optical depth (AOD) and Angström exponent (ANG). Calculations are performed for the whole month of July 2016 and are based on hourly data. The bias is computed as model minus observation.*

Results show that the spatial correlation has globally high values (between 0.44 and 0.97), showing that the horizontal gradients of meteorological variables or aerosols are correctly modelled. The 2m-temperature is probably the less meaningful indicator despite very good results, the diurnal cycle and the latitudinal effect having an important weight in the estimation of this parameter. It is not the case for the wind speed for which the spatial correlation of $R_s = 0.79$ shows that the model is able

to reproduce the regions characterized by low and high wind speeds. On the other hand, the temporal correlation $R_t = 0.25$ shows that the diurnal cycle of the wind speed could be improved. A positive bias is diagnosed, bias=1.26, and this could induce a slight overestimation of transport close to the surface as well as mineral dust emissions. Regarding the total precipitations, the spatial correlation, $R_s$=0.64 is rather high showing that the clouds front of the monsoon is well modelled. However, the temporal correlation, $R_t$=-0.09, is very low and shows that precipitations are not modelled at the right time. In addition, a

negative bias of 0.21, is calculated, showing that the WRF model under-estimates the precipitation for this period and region. Finally, the statistical scores for the aerosols, AOD and (ANG, have values showing that the modelling of aerosols is correct. For the spatial correlation, we obtain $R_s$=0.44 and $R_s$=0.97 for AOD and ANG respectively, meaning that the relative part of fine/coarse particles is well modelled, but the plumes may be misplaced. The bias is positive for AOD (bias=0.14) and negative for ANG (bias=-0.17) meaning that there is too much aerosol with the diameter corresponding to the optically active





wavelength of the AOD calculation and it also corresponds to the negative bias, showing there is too much fine compared to coarse particles.

## 5  Impact of emissions scenarios

Results are presented as monthly map of averaged model results for July 2016. All variables are bidimensional except the surface concentrations of ozone and $PM_{2.5}$ (the map represents the concentrations at the first model vertical level, i.e between

surface and 20 m AGL). For each variable, results are presented as CPLfull values and as values of differences (CPLanthro-CPLfull) and (CPLdust-CPLfull).

### 5.1  Significance of differences

To quantify the statistical significance of differences, the Mann-Whitney test is applied (also called Wilcoxon signed rank test), (von Storch and Zwiers, 2001). This test is non parametric: there is less restrictive assumptions as, for example, the fact that the

distribution has not to be normal. This test examines the two sets of data (in our case, two different simulations) by combining all data and by sorting them in ascending order.

The first and second simulations, noted $x_{1,i}$ and $x_{2,i}$, have here the same dimension, $N$. We first calculate the difference between the two datasets:

$$d_i = x_{2,i} - x_{1,i} \ , \text{For } i = 1, ..., N \tag{7}$$

A reduced dataset with dimension $N_r$ is built by removing data where $d_i$=0. The remaining data $d_i$ are sorting, in ascending order and their rank, $R_i$, is stored. The statistic test $W$ is calculated as:

$$W = \sum_{i=1}^{N_r} (sign(d_i) \times R_i) \tag{8}$$

A $z$ score is the calculated as:

$$z = \frac{W}{\sigma_w} \tag{9}$$

with, when $N_r \geq 20$:

$$\sigma_w = \sqrt{\frac{N_r(N_r + 1)(2N_r + 1)}{6}} \tag{10}$$

The null hypothesis $H_0$ is rejected (i.e the differences are significant and not due to hazard) if $|z| > z_{critical}$. For a level of significance of 0.05, we have the value $z_{critical} \approx 1.645$. In the following, Figures present crosses at the points where the difference was found to be significant.



CPLfull                    (CPLanthro-CPLfull)                    (CPLdust-CPLfull)

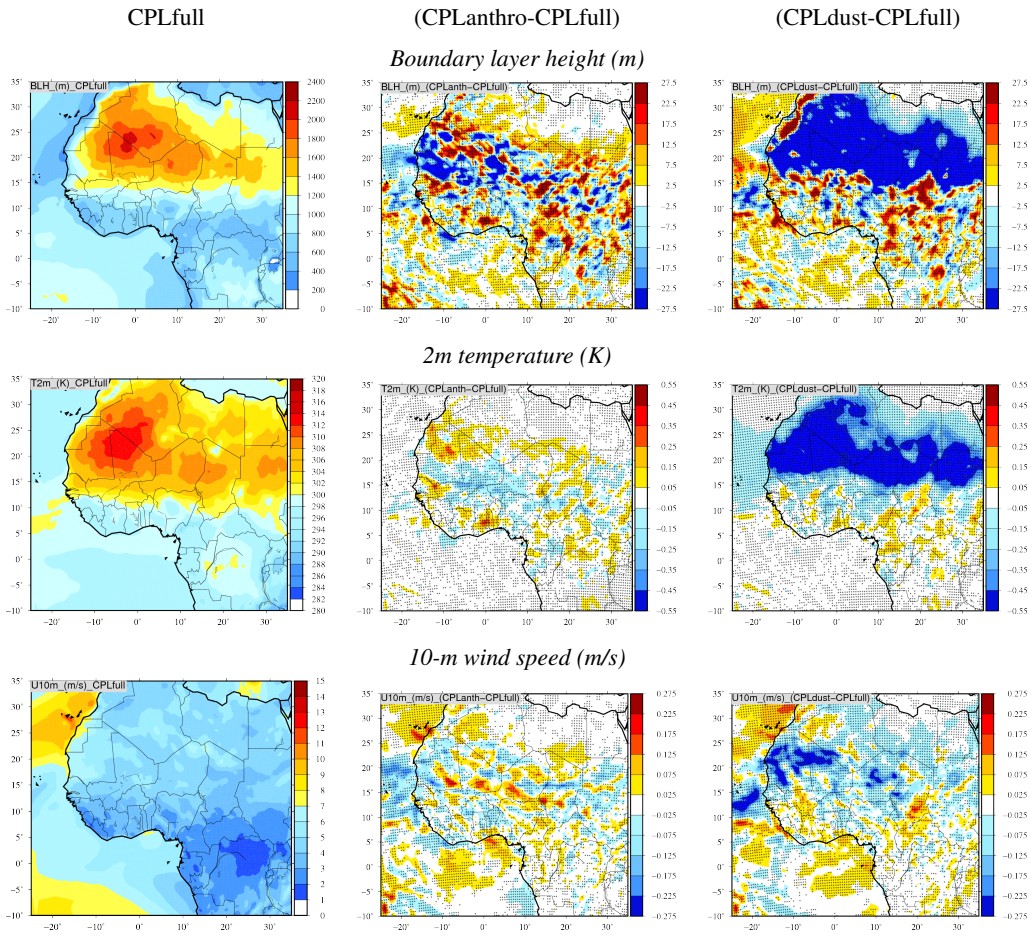

**Figure 5.** *Left column (from top to bottom): Maps of monthly mean boundary layer height (m), 2-m temperature (K) and 10-m wind speed (m s⁻¹) for July 2016. Middle column: maps of monthly mean of daily averaged differences between CPLanthro and CPLfull for same variables as the left row. Right row: maps of monthly mean of daily averaged differences between CPLdust and CPLfull for same variables as the left row.*

## 5.2 Meteorology

Results for boundary layer height, 2-m temperature and 10-m wind speed are presented in Figure 5. The boundary layer height (BLH) is higher over land, more specifically over desert areas. Due to the cooler monsoon, the BLH is lower than 1000 m over sea and for latitude $< 15^{o}$N and may reach $\approx 2000$ m over the Sahara, and even greater depth in the region of the heat low (recall that these are daily averaged values). The differences range in the interval $\pm 30$ m. Values are alternatively positive and negative. There is no mean tendency, except over the Saharan desert where the differences are always negative in case of the





CPLdust Vs CPLfull comparison, indicating that the response is related to the direct effect of aerosols and the reduction of 2-m temperature.

As BLH, the 2-m temperature is higher over land than over sea and the largest values are modelled over the Saharan desert. For the differences (CPLanthro-CPLfull), 2-m temperature differences are small, at the maximum $\pm 0.5$ K. They are negligible
over the sea and positive along the coast, then negative for latitude $\approx 15^o$N, then positive for latitude up to $\approx 20^o$N. For CPLdust, the impact is negative north of $\approx 15^o$N, i.e where mineral dusts are present. Reducing dust emissions leads to a decrease of 2-m temperature. This result is close to the ones of Han et al. (2013) who showed a decrease in temperature and wind speed during the daytime, and the opposite effect during the night, when the dominant aerosol is mineral dust.

The 10-m wind speed is lower over land than over sea in average. The differences between the different simulations are
very variable in space. For CPLanthro, there is some increase of the wind speed in the desert which is not present in CPLdust. Interesting, we are able to identify a line of enhanced wind speed spanning from the southeast corner of Mauritania to the southern part of Chad, across the Sahel. This feature is seen in the same region where the 2-m temperature is consistently negative in the CPLanthro-CPLfull differences. As for temperature and depending on the location of the dust plume, the impact is different on wind speed: as shown by Miller et al. (2004), and confirmed by Rémy et al. (2015), the temperature
decreases under the plume, increasing atmospheric stability and reducing the wind speed. But, at the edge of the dust plume, a horizontal thermal gradient is more pronounced, leading locally to an increase of the surface wind speed. Thus, depending on the location of the dense mineral dust plume, it is expected to have increase or decrease of the 10-m wind speed.

These results first show that, for all locations where values are non-negligible, results are all significant. Second, even if emissions (anthropogenic or dust) are located in specific areas only, the impact of their changes affect the whole simulated
domain. The differences are very patchy and alternate between negative and positive values. The differences values are not very high and represent a few percent only for each variable.

### 5.3    Rain and radiation

Results are presented in Figure 6. The rain mixing ratio is first presented. This corresponds to the first vertical model level of the rain profile minus the evaporation: it corresponds to the amount of water finally reaching the surface. It is non-zero mainly
between the latitudes +5/+15 $^o$N, representing the monsoon during this period. The most important values are around +15 $^o$N and the differences tend to zero up to this latitude. Interestingly, July monthly precipitation shows two distinct bands around 10 $^o$N and 15 $^o$N respectively, indicating that the WAM is not yet fully developed in the Sahel (see Knippertz et al. (2017)). The two scenarios simulations provide similar kind of differences: the values are alternatively positive and negative showing that the precipitation front is moving northward. Differences are in the range $\pm 20$ kg/kg, representing 25% of the maximum
modeled values. All calculated differences are significant. It is worth noting that in the monthly averaged CPLanthro-CPLfull difference, the enhanced linear wind speed feature seen in Figure 5 is located to the north of the enhanced linear rain feature. Such a clear match between the positive rain and wind speed anomaly is not seen in the CPLdust-CPLfull differences.

For radiation, we compare the effects of the coupling on shortwave (SWsurf) and longwave (LWsurf) radiation net fluxes at the surface. SWsurf and Lwsurf are estimated by subtracting the upwelling from the downwelling flux at the surface. The



CPLfull  (CPLanthro-CPLfull)  (CPLdust-CPLfull)

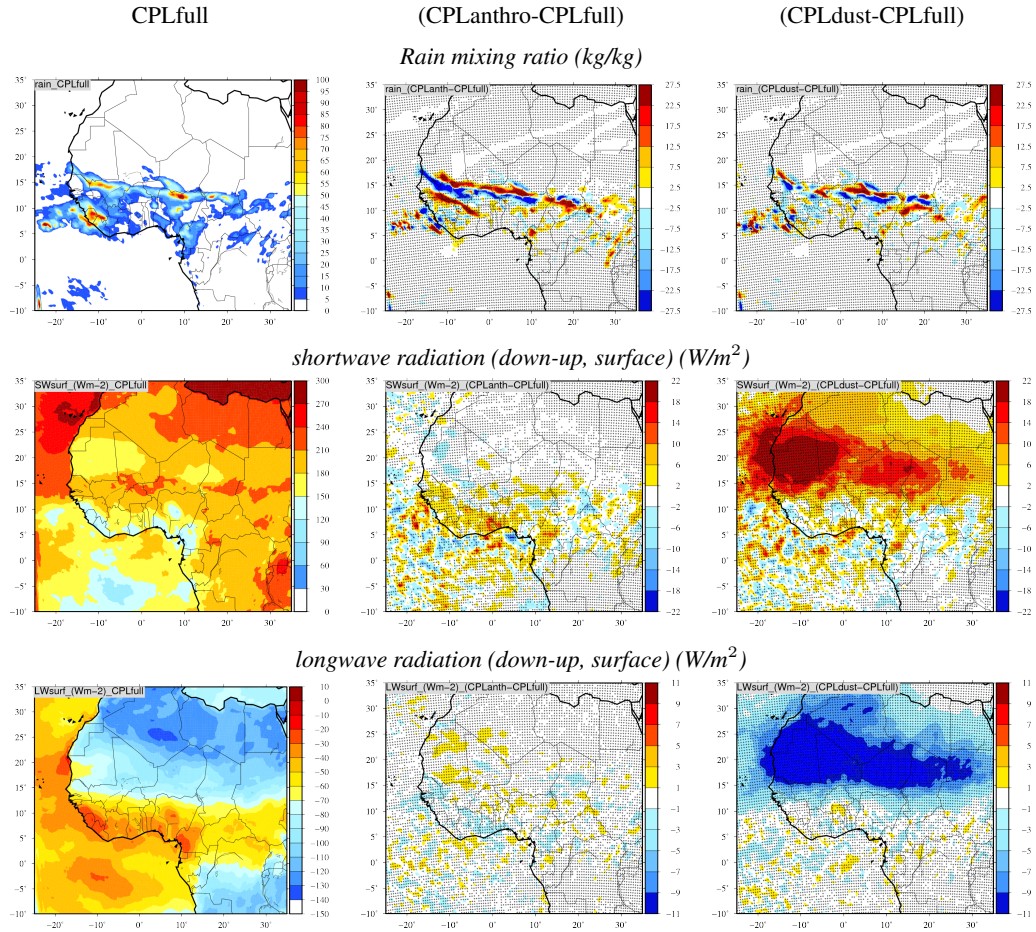

**Figure 6.** *Left column (from top to bottom): Maps of monthly mean rain mixing ratio (kg/kg), shortwave radiation (down-up, surface) (W/m$^2$) and longwave radiation (down-up, surface) (W/m$^2$) for July 2016. Middle column: maps of monthly mean of daily averaged differences between CPLanthro and CPLfull for same variables as the left row. Right row: maps of monthly mean of daily averaged differences between CPLdust and CPLfull for same variables as the left row.*

coupling shows low effect for CPLanthro, with SWsurf and LWsurf differences showing an alternance of negative and positive values over most of the domain. However, there is an indication of consistent enhancement of SWsurf over the Sudanian region bordering the Gulf of Guinea between Senegal and Nigeria, i.e. south of the rain band displaced northward in CPLanthro. In the case for CPLdust, where the decrease of dust emissions induces an increase of SWsurf and a decrease of LWsurf over the Saharan region. Nevertheless, in the rest of the domain, the differences in SWsurf and LWsurf exhibits a noisy behavior.

At the maximum, the increase of SWsurf may reach $\approx$ +20 W/m$^2$, representing 10% of the maximum flux (averaged over the month). The change is also $\approx$ 10% for LWsurf. This aerosol effect was already presented, among other regions, over Australia in Choobari et al. (2013) and over Europe in Bangert et al. (2012). It was also discussed over Africa in Briant et al.





(2017). Depending on the size distribution, mineral dust may absorb or scatter the radiation. During the day, aerosol absorbs the shortwave (i.e solar) radiation, inducing a heating of the atmosphere, a cooling at the surface, and a decrease in the cloud
coverage. During the night, aerosol increase induces a longwave radiation increase, then an increase in temperature close to the surface.

## 5.4 Atmospheric composition

Finally, results for mineral dust emissions, surface concentrations of $O_3$ and $PM_{2.5}$ and AOD are presented in Figure 7. On average, surface ozone concentrations are the most important over the sea. Low values are modeled for latitudes between +5
and +15$^o$N, showing the impact of precipitation and low short wave radiation on ozone production and deposition. The two emissions scenarios have completely different impacts on ozone. The CPLanthro shows that reducing anthropogenic emissions would decrease ozone over land, mainly due to a decrease of available reactive VOCs. But it also increases ozone over sea and it could be due to a change in radiation and cloud cover change, as pointed out by Forkel et al. (2012) over Europe. For CPLdust, less dust emissions lead to less AOD, then more radiation thus more photochemistry. The variability on ozone is $\approx 3$
$\mu$g m$^{-3}$, corresponding to $\approx 5\%$ of the maximum of the monthly averaged concentration. All calculated values are significant.

The second result concerns the impact of ARCI on $PM_{2.5}$. For this figure, we have chosen to present $PM_{2.5}$ without the mineral dust contribution. These fine particles are then representative of contributions from anthropogenic and biogenic emissions only. For CPLanthro, less anthropogenic emissions lead to less surface $PM_{2.5}$ concentrations, essentially where the particles are emitted. For CPLdust, some positive and negative differences are present over Central Africa. It may be complex feedbacks
loop between radiation, meteorology and emissions, difficult to identify. The values are low but remain significant.

For mineral dust emissions, the differences evidence very interesting results. First, the map of emissions shows their location: up to 15$^o$N in latitude, northward to the precipitation identified previously. The scenario of CPLdust on mineral dust gives a linear impact: less emissions are visible on the map of difference, then less surface concentrations, then less AOD over the Sahara. Non-zero differences are of course spatially limited to the area where dust are emitted. More surprising, the scenario
of CPLanthro has an impact on mineral dust emissions. Emissions decrease close to the coast, leading to more mineral dust concentrations over the Sahara.

With the scenario CPLanthro, the increase of mineral dust emissions in the Sahara leads to more AOD. The impact is not very important (+0.1) but remains significant. The enhanced dust emissions are seen to coincide with the enhanced linear 10-m wind feature ahead of the northward displaced rain band seen in Figure 5 and Figure 6. A more thorough explanation is
proposed in the next section.



**Figure 7.** *Left column (from top to bottom): Maps of monthly mean of ozone (ppb), PM$_{2.5}$ (μg/m$^3$) (without mineral dust), mineral dust emissions (g m$^{-2}$ h$^{-1}$) and Aerosol Optical depth for July 2016. Middle column: maps of monthly mean of daily averaged differences between CPLanthro and CPLfull for same variables as the left row. Right row: maps of monthly mean of daily averaged differences between CPLdust and CPLfull for same variables as the left row.*





## 6 Focus on the mineral dust emissions

### 6.1 Correlations between differences

As displayed in Figure 7 and for the scenario on anthropogenic emissions, the map of differences for mineral dust emissions and AOD shows a significant increase in the Sahara. Knowing there is no significant anthropogenic emissions in this area,

there is no reason to directly increase PM and AOD in this region. The only possible reason is a change in the meteorology, propagating across the domain and impacting the mineral dust emissions. The explanation probably comes from the changes in 10-m wind speed as shown in Figure 5: a decrease in emissions leads to an increase of wind speed over the Sahara.

   The fact that changes in wind speed leads to an increase of mineral dust emissions is likely related to the non-linear relation between wind and mineral dust in the physics of the emissions fluxes. These emissions depend on a threshold value for wind

speed. Up to this value they increase exponentially with wind speed. Wind Gusts are more efficient at producing large dust emission than a steady flow for which the mean speed is just above the velocity threshold.

   The impact of an increase of wind speed is a good candidate to explain the increase in dust emissions. In order to prove this, we performed spatio-temporal correlation calculations between the change in dust emissions and the change in 10-m wind speed, boundary layer height, 2-m temperature and total precipitation for each of the scenario, based on the maps of differences

shown in Figure 5. Correlations are computed for each hourly output of the simulation and for the whole month of July 2016. They are presented in Table 2.

| | CPLanthro -CPLfull | CPLdust -CPLfull |
|---|---|---|
| Variable | E(dust) | E(dust) |
| U10m | 0.81 | 0.62 |
| PBLH | 0.15 | -0.01 |
| T2m | -0.12 | -0.27 |
| topc | -0.14 | -0.08 |

**Table 2.** *Monthly averaged correlations between changes in dust emissions and changes in 10-m wind speed (U10m), boundary layer height (PBLH), 2-m temperature (T2m) and total precipitation (topc).*

   The best correlation between a meteorological parameter and mineral dust emissions is for 10-m wind speed. For emissions reductions CPLanthro, a high correlation of 0.81 is calculated. The correlation for CPLdust is less important with a value of 0.62. The reason is that the decrease of mineral dust emissions is here not due to a meteorological parameter but to the

scenario choice itself. The correlations between differences in emissions and differences in PBLH, T2m and precipitation are low, meaning that their changes are not the cause of mineral dust emissions changes.





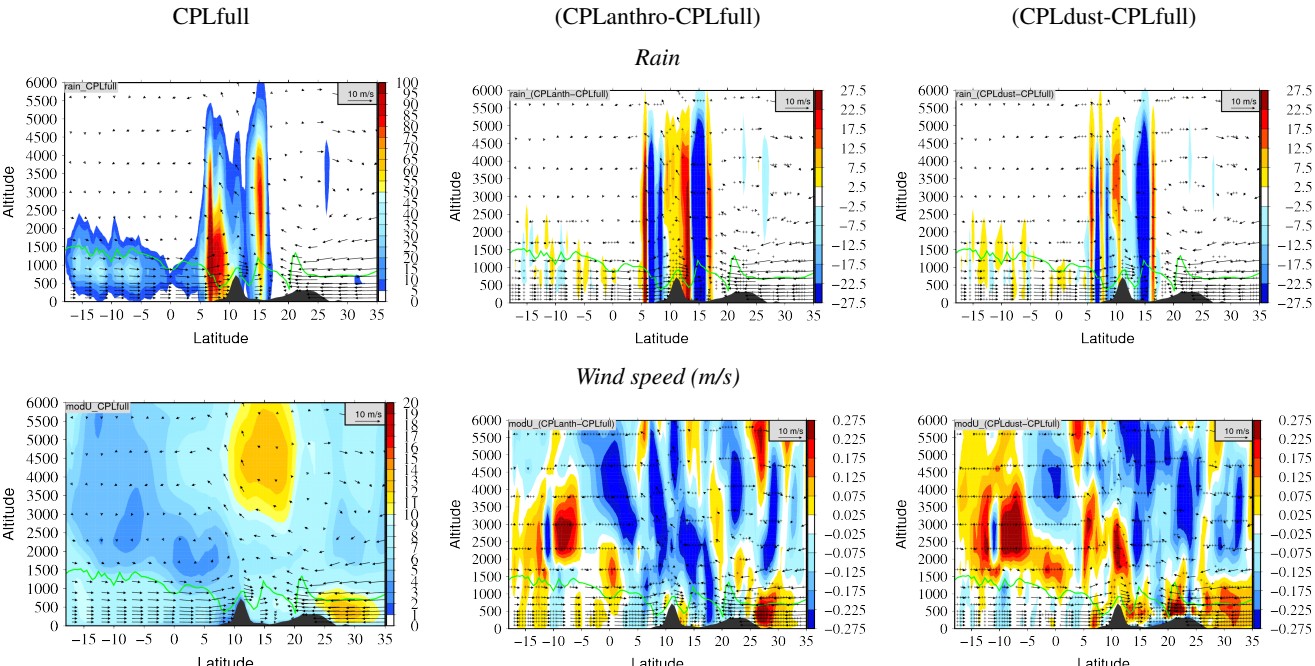

**Figure 8.** *Vertical cross-section of rain and wind speed vertical profiles, averaged from 1 to 31 July 2016 and for longitude -10$^o$. The mean value for CPLfull and the differences (CPLanthro-CPLfull) and (CPLdust-CPLfull) are presented. Crosses indicate locations where the differences are significant.*

## 6.2 Vertical cross-sections

In order to better understand the impact of anthropogenic emissions, we analyse in Figure 8 the monthly averaged vertical cross-sections of rain and wind speed (averaged between -12$^o$ and -8$^o$).

The rain is mainly located over the sea and over land at two distinct locations, in a band between 5 and 10$^o$N and around 15$^o$N, respectively. For the differences, the most important values are also highlighted at these latitudes. Dipoles of negative and positive differences appear where the reference simulation is showing maxima at these latitudes. It means that the maxima of rain were shifted in latitude. Viewing the locations of the maxima and the differences, it is clear that the rain band located at 15$^o$N moved northward in the two simulations.

The mean wind speed along the north-south transect highlights the presence of the AEJ at 15$^o$N between 3500 and 5500 m AGL. Close to the surface, maximum wind speeds are seen between 25 and 30$^o$N, over northern Africa. the wind speed increases in this region, in both sensitivity experiments so that reducing emissions tends to increase wind speed in the northern part of the domain. In the CPLdust simulation, the wind speed is also increased over the Sahara. For both simulations, we see an increase in near surface wind speed where the shift of rain band is most pronounced. In altitude, over the continent, the

difference fields are quite noisy, but suggests an northward shift of the AEJ in CPLanthro and a southward shift of the AEJ in





CPLdust. Nevertheless, monthly the averaged cross-sections of differences between the reference simulation and the sensitivity experiments are difficult to interpret: in the next section, we will thus represent the data as Hovmöller diagrams, (Hovmoller, 1949).

## 6.3 Hovmöller diagrams

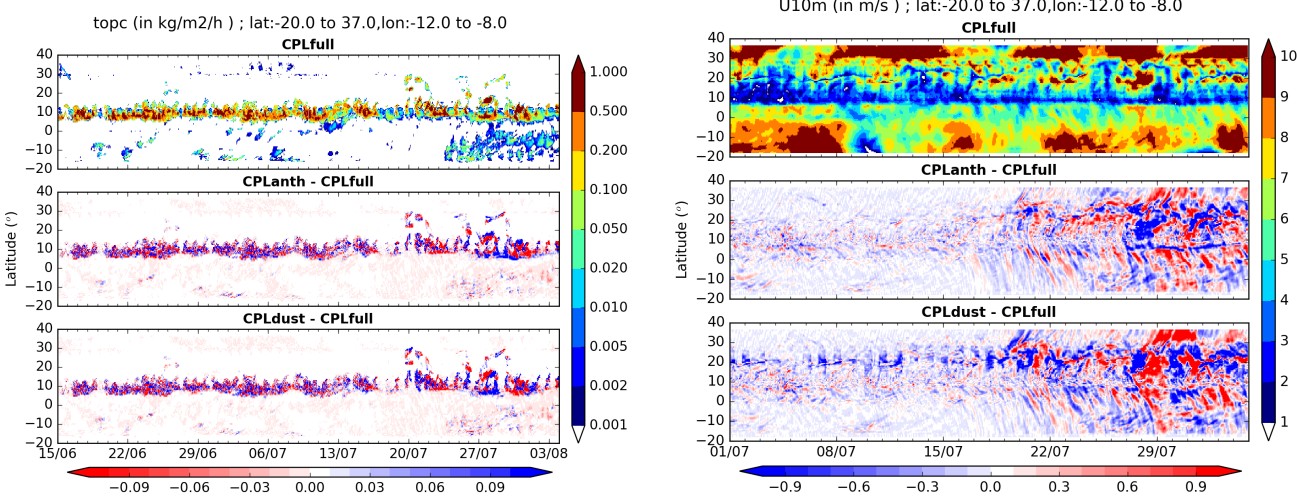

**Figure 9.** *Hovmöller diagrams for total precipitation and 10-m wind speed.*

Hovmöller diagrams are presented for the total precipitation and 10-m wind speed in Figure 9. Temporally, the data are daily averaged. Spatially, they are averaged in longitude between -12 and -8° as for the vertical cross section previously presented. The main area with precipitation, between 10 and 15°N, is clearly shown. The progression in time of precipitation shows that in July 2016 the coastal phase of the monsoon still coexists with the Sahelian phase, (Knippertz et al., 2017). In particular, an intense event in the Sahel is simulated on the 27 July, synchronous with an intense counterpart around 5°N. Moreover, on 385 30-31 July, a breakdown of the Sahelian phase is also simulated. Additional moderate precipitations between -10 and 0°N appear after 20 July only for a few days, that do not appear on the monthly averaged maps, Figure 6. They are not due or influenced by the ARCI since they are not visible in the Hovmöller diagrams of differences. These differences have variability (with alternate negative and positive values) mainly when the precipitation are the more important, for latitude between 10 and 15°N, all along the period. There is no significant difference between the two scenarios, CPLanthro and CPLdust.

Results are completely different for the 10-m wind speed. The time evolution for CPLfull shows several regimes but the most important values are always for latitude corresponding to the sea (between -20 and -10°N) and over the Sahara (between 25 and 40°N), while the weakest winds are seen at the coast, and up to 15°N, on average, in spite of a few episodes of strong winds around 18, 22 and 27 July. The diagrams of the differences show that the main differences are at the end of the studied period, after 15 July 2016. As for other variables, the negative and positive values alternate depending on the latitude.





Nevertheless, there is a coherent positive 10-m wind feature around $17^o$N between 21 and 27 July, that is consistent with the horizontal wind anomaly shown in Figure 5. The linear structure of the positive and negative anomalies north of the coastline also suggest a northward displacement of the wind anomalies with time. Similar propagating anomalous features are seen in CPLdust, with the positive anomalies being more pronounced that in CPLanthro. For instance, the strong wind episode seen around $20^o$N in CPLfull is enhanced on CPLdust, as opposed to CPLanthro in which it is damped. It shows that the ARCI

change the atmospheric flow during the latter part of the simulated period and most of the domain, including the ocean where wind anomalies are seen to propagate southward. The impact of the ARCI on the wind speed is more important for the scenario CPLdust than for CPLanthro. It is linked to the fact that the scenario CPLdust reduces much more the content of aerosols in the boundary layer and troposphere than CPLanthro. Nevertheless, the link between the anomalous precipitation, surface net shortwave flux, 10-m wind speed and dust emissions are more coherent et more enhanced in CLPanthro.

**7 Conclusions**

The months of June-July 2016 were modelled using the WRF-CHIMERE regional models over a large domain centered on the Gulf of Guinea. The modelled period corresponds to the DACCIWA Intensive Observation Periods. The modeling system was used with the addition of the meteorology/aerosol coupling to describe the aerosol direct and indirect effects. The model outputs from a reference simulation were first compared to surface observations. It was shown that the model represents the

meteorology and the aerosols concentrations in a realistic manner.

Two scenarios were used to compute additional simulations with halved emissions of mineral dust and anthropogenic sources. By comparison between the reference case and these scenarios, the direct and indirect effects of aerosol were quantified. Overall, results show moderate impact of the direct and indirect effects, as also quantified over Europe in Forkel et al. (2015). For each meteorological parameter, the impact represents a few percent of the monthly mean value. Furthermore, the

direct and indirect effects appear to be increasing with time.

Due to the larger amount of aerosol injected in the troposphere, the mineral dust scenario induces the most important changes in the meteorology. The scenario of anthropogenic emissions reduction leads to patchy impacts with alternating positive and negative changes in the maps of differences between the scenario and the reference simulation outputs. On the other hand, the mineral dust are at the origin of important changes, homogeneous and over the whole Saharan region.

A surprising feedback was identified with these scenarios. When anthropogenic emissions are reduced along the Gulf of Guinea, the precipitation front moves northward and the 10-m wind speed increases in the Sahel and Sahara regions. Consequently, the mineral dust emissions are enhanced, leading to more important surface concentrations of aerosols, then enhanced AOD. By changing the meteorology and the precipitations, a decrease of anthropogenic emissions would increase mineral dust several thousands of kilometers northward. These changes are not very important but are statistically significant. One also

has to note that anthropogenic emissions evolve and increase quickly in the Gulfe of Guinea and simulations were probably performed with underestimated anthropogenic emissions. About the feedback of the emissions reduction linked to a dust emissions increase, the same mechanism also applies for the dust scenario. But this scenario being itself about an important dust





emissions reduction, the potential increase due to wind speed change is completely masked by the emissions scenario leading to a decrease.

*Author contributions.* LM designed the experiments, performed the simulations. PT and LM developed the model code. LM prepared the manuscript with contributions from all co-authors for the analysis of the results, the preparation of the figures and the manuscript.

*Competing interests.* The authors declare that they have no conflict of interest.



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
