# Peer review of "The role of aerosol-radiation-cloud interactions in linking anthropogenic pollution over southern West Africa and dust emission over the Sahara"

_Atmospheric Chemistry and Physics, 2019_

## Referee Comment (RC1) · Anonymous Referee #1 · 3 Oct 2019

Menut et al. present a regional modelling study to evaluate the direct and indirect radiative effects of mineral dust and anthropogenic air pollution particles over West Africa for the period of the DACCIWA field campaign in July 2016. The simulations were performed with the model system WRF-CHIMERE allowing for online interaction of aerosol particles with radiation and clouds. Standard meteorological parameters, soundings and measurements of aerosol optical depth were used for model evaluation. The analysis shows that air pollution over southern West Africa appears to influence dust production in the Saharan Desert through direct and indirect aerosol radiative effects. Their study is an interesting contribution to the topic, in particular as it shows how natural and anthropogenic aerosols are interdependent and can influence the climate.

I recommend to publish this work in Atmospheric Chemistry and Physics, however with the following comments being considered.

General comments:

In my opinion, the goal of the study to evaluate the direct and indirect aerosol effects over West Africa was not fully achieved. This would need to disentangle aerosol-radiation and aerosol-cloud interactions, which however requires more than two sensitivity model runs. Direct radiative effects and rapid adjustments seem to dominate the aerosol impact on boundary layer properties, precipitation and atmospheric composition, including mineral dust (also in the way the results are presented). Therefore, I also wonder whether the title is in accordance with the text.

Considering the fact that this paper is to be published in the DACCIWA special issue, the authors should not miss the opportunity to use the rich dataset for a detailed evaluation, in particular of the aerosol-cloud interactions. In addition, nowadays, the computational costs for a 60-km regional simulation should allow for more than 1.5 months. Could you imagine to extend your model runs to cover the whole summer season or even one year?

Language wise, the manuscript is already in a good state. The text could nonetheless use another round of editing to eliminate remaining minor inconsistencies and typos. Throughout the manuscript, the format of references has to be revised.

Specific comments:

1. Page 1, lines 8-10: It should be mentioned that the values of modelled aerosol effects on temperature and radiative fluxes are monthly averages for July 2016.

2. Page 1, lines 12-13: In the Abstract but also in the Results' part and Conclusions, the impact of dust and anthropogenic air pollutants on the wind field and precipitation is presented rather descriptively. So, it remains unclear, what the connection is between the latitudinal shift of the monsoonal precipitation and an increase in surface winds,

in particular, since moist convective cold pools are not resolved in the 60-km model simulations. Here, the authors could strive for a more thorough explanation, possibly, in the context of the West African Monsoon circulation.

3. Pages 2, lines 25-34: It might be worth mentioning that also Heinold et al. (2011, doi:10.1111/j.1600-0889.2011.00574.x) had already found very similar effects of Saharan dust and biomass burning smoke strengthening the Hadley circulation, which influenced the aerosol distribution in a similar way as described in this study.

4. Page 2, line 35: Aerosol particles involved in heterogeneous freezing, today, are more commonly referred to as "ice nucleating particles (INP)".

5. Pages 5/6, Section 3.2: Does the radiation scheme in the model consider the change in cloud properties due to the aerosol-cloud interactions?

6. Page 6, Section 4: To my knowledge, extensive aircraft measurements of aerosol chemistry, radiation, and cloud-aerosol interactions took place during DACCIWA. Why were these observations not used for model evaluation?

7. Page 21, lines 17-18: The alternating patterns are most likely due to stochastic effects of clouds between the two model representations.

8. Fig. 4: The different coloured lines are too thin and hard to distinguish. Since the individual soundings are not discussed anyway in the text, I wonder whether it would make sense to average over the profiles (or groups of them).

9. Figs. 5 to 7: In the difference plots, the tiny black dots probably indicate statistical significance. This should be mentioned in the figure caption.

10. Figs. 4 to 8: The font size of axis labels and titles and/or colour bars is too small and needs to be adjusted.

---

## Referee Comment (RC2) · Anonymous Referee #2 · 4 Oct 2019

The paper entitled "The role of aerosol-cloud interactions in linking anthropogenic pollution over southern West Africa and dust emission over the Sahara" studies the aerosol direct and indirect effects over West Africa during the DACCIWA in July 2016 using the WRF-CHIMERE coupled model. A reference case is compared against two scenarios with halved emissions of mineral dust and anthropogenic sources, obtaining significant results, even though the impact of the direct and indirect effects is moderate. The paper is a significant contribution to the field and the obtained results are of interest. The paper is well written and the structure is clear. My recommendation is publication after minor revisions.

[Figure]

General comments:

The results section is slightly descriptive and a deeper discussion of the results is missing at some points, especially in Section 4 where the modelled data are compared to observations. How does the differences observed here between the model and the data affect the results of the study? What are the uncertainties? Additionally, it is necessary to revise the whole manuscript for typos, paying special attention to the references format.

Specific comments:

Page 6, Line 160: Given the importance of biomass burning aerosols, as explained by the authors in the introduction, why is it not included in the analysis?

Page 9, Line 240: Is there any possible explanation for this bias in Savè?

Figure 8: The crosses and wind arrows are difficult to distinguish. Please improve the readability of the figure.

Page 21, line 414: Could you provide a quantitative estimate of this percentage?

Page 21, lines 414-415: "Furthermore, the direct and indirect effects appear to be increasing with time." It is not clear to me how you reach to this conclusion. Please, explain.
* * *

---

## Author Comment (AC1) · 6 Nov 2019

**The role of aerosol-cloud interactions in linking anthropogenic pollution over southern West Africa and dust emission over the Sahara**

Laurent Menut et al.
https://doi.org/10.5194/acp-2019-658

Dear Editor and reviewers,

We acknowledge the reviewers for the time spent to evaluate our work and for their minor revisions. We also acknowledge the Editor and we made all proposed changes in the revised manuscript. Please note that answers are in blue and after each reviewer's remark.

All reviewers remarks were taken into account and are detailed in this letter. Summarizing our answers:    10

1. Text, references and Figures (captions and labels) were checked and corrected as requested.

2. A more precise and physical explanation was added to interpret why a change in aerosols near the coast impacts the dust emissions. This is achieved by adding the calculation of maps and vertical cross-sections of Moist Static Energy (MSE).

3. Several references were added in the manuscript: Heinold et al. (2011), Slingo (1989), Stephens et al. (1990), Fontaine   15 and Philippon (2000), Sultan and Janicot (2003), Neelin and Held (1987).

4. A comparison to DACCIWA flight measurements was added to better analyze the reference simulation before the analysis of the scenarios impact. It shows mainly that the low resolution causes smoothed latitudinal variations compared to high frequency aircraft measurements. But, the most important (regarding the specific focus of this study) is that mean averaged values are comparable between model and measurements, as for the other measurements   20 already in the submitted manuscript. Then, the scenario study is able to provide correct order of magnitude for the calculated changes.

Best regards,
Laurent Menut
November 5, 2019

**1    Reviewer #1**

Menut et al. present a regional modelling study to evaluate the direct and indirect radiative effects of mineral dust and anthropogenic air pollution particles over West Africa for the period of the DACCIWA field campaign in July 2016. The simulations were performed with the model system WRF-CHIMERE allowing for online interaction of aerosol particles with radiation and clouds. Standard meteorological parameters, soundings and measurements of aerosol optical depth were used for model evaluation. The analysis shows that air pollution over southern West Africa appears to influence dust production in the Saharan Desert through direct and indirect aerosol radiative effects. Their study is an interesting contribution to the topic, in particular as it shows how natural and anthropogenic aerosols are interdependent and can influence the climate.

I recommend to publish this work in Atmospheric Chemistry and Physics, however with the following comments being considered.

**General comments:**

In my opinion, the goal of the study to evaluate the direct and indirect aerosol effects over West Africa was not fully achieved. This would need to disentangle aerosol-radiation and aerosol-cloud interactions, which however requires more than two sensitivity model runs. Direct radiative effects and rapid adjustments seem to dominate the aerosol impact on boundary layer properties, precipitation and atmospheric composition, including mineral dust (also in the way the results are presented). Therefore, I also wonder whether the title is in accordance with the text.

*Answer:*

It is logical to ask for disentangle the direct and indirect effects. In a first unpublished version of this study, results were presented like that. We presented 4 simulations: no interactions with aerosols (called noCPL), only with direct effects (CPLdir), only with indirect effects (CPLind) and with the direct+indirect effects (CPLfull). Results showed that the main effects are due to direct effects. But apart this conclusion, there was no really new interesting results. And it was not realistic since this is not possible to split these contributions in real conditions. In addition, there are some interactions between the two and to make simulations with only one can not be correct.

It is why we preferred to calculate the two effects together (reality) and to quantify the impact of emissions reduction (realistic) in place of processes one by one (not realistic).

For the title, the reviewer is right if he thinks that we forgot to focus on the main effect: we forgot the word 'radiation' in the title and we added it in this new version. The title is thus now *The role of aerosol-radiation-cloud interactions in linking anthropogenic pollution over southern West Africa and dust emission over the Sahara.*

Considering the fact that this paper is to be published in the DACCIWA special issue, the authors should not miss the opportunity to use the rich dataset for a detailed evaluation, in particular of the aerosol-cloud interactions. In addition, nowadays, the computational costs for a 60-km regional simulation should allow for more than 1.5 months. Could you imagine to extend your model runs to cover the whole summer season or even one year?

*Answer:*

Yes, this is a good point: DACCIWA offers a lot of measurements. But, for the specific case of this study, we want to highlight that study is more a model vs model study, with scenarios of emissions. This is why, for the model validation, we focussed on measurements with a large spatial extent. In general, it is the case for model grids with a low resolution such as this one.

By principle, comparison with measurements, such as DACCIWA aircraft measurements, is correct for simulations with horizontal resolutions closer to the km or a few km. With $\Delta x$=60km, there is only 2 or 3 model cells to represent a whole aircraft flight: thus, we can compare something but the interest is very limited.

However, to comply with this request, we propose here some comparisons to aircraft measurements. We selected two flights conducted during the 13th July 2016, with the German Falcon aircraft and the French ATR aircraft. Results are presented in Figure 1 . This new Figure and the following paragraph were added in the revised manuscript.

In the manuscript:

*Figure 1 presents the comparison between aircraft measurements and the model results with the reference case (CPLfull) and the two scenarios (CPLanthro and CPLdust). The two flights were operated during the 13th July 2016, from 09:00 to 12:00 UTC for the German DLR Falcon aircraft and from 12:00 to 15:00 for the French ATR aircraft. Details about these flights are presented in Flamant et al. (2018). These flights were selected because: (i) they were done at constant longitude and are thus well designed to discuss the monsoon latitudinal behaviour, (ii) they are among the lastest flights performed during the campaign and thus close to the second part of July when several monsoon phases were identified, (Knippertz*

[Figure]

**Figure 1.** *Model versus aircraft measurements. Flight of 13th July 2016 with the DLR (left) and ATR (right) aircrafts. The top Figures show the altitude (AGL) and the latitude during the flight. The other panels present the concentrations of CO (ppb), NO$_2$ (ppb) and O$_3$ (ppb).*

*et al., 2017). Results show that the background values are well modelled for CO, NO$_2$ and O$_3$. The comparison failed when observations show a large temporal variability: in this case, the coarse model resolution, not designed to make such type of comparison, show its limits and the model is not able to retrieve this high variability. For these species, it is also shown that the three simulations, including the two scenarios, gives close values, showing that, in altitude, scenarios have a lower impact than close to the surface.*

About the model runs extension: It is possible but we are not convinced of the benefit of a longer period. We focused on the DACCIWA Intensive Observation Periods, in order to link the simulations to all other papers analyzing this period. The goal of the paper is to estimate the impact of aerosols effects before and after the monsoon onset. We can add 2 or 3 months of simulation, but the results will probably be of the same kind. And in addition, the reviewer is probably too much optimistic about our model. We don't know the speed of WRF-chem or COSMO-MUSCAT, for example, but in the case of WRF-CHIMERE, it is about 6 days to run the 1.5 months. Multiplied by 2 to have 3 months and by 3 for the three configurations, it is more than 36 days of simulations (and considering that the runs will be made without any crash, and, obviously, it is never the case). Considering that we certainly won't learn more new things and that 1.5 months is enough to stabilize all atmospheric processes, we prefer to keep the configuration used for this article. For the next one, we will think about the Reviewer #1 point of view and run more time.

Language wise, the manuscript is already in a good state. The text could nonetheless use another round of editing to eliminate remaining minor inconsistencies and typos. Throughout the manuscript, the format of references has to be revised.

*Answer:*
All references were checked during the technical phase of the submission. Indeed, some errors remained and were corrected in this new manuscript version.

**Specific comments:**

1. Page 1, lines 8-10: It should be mentioned that the values of modelled aerosol effects on temperature and radiative fluxes are monthly averages for July 2016.

   *Answer:*
   Yes, this is correct and it was corrected.

2. Page 1, lines 12-13: In the Abstract but also in the Results' part and Conclusions, the impact of dust and anthropogenic air pollutants on the wind field and precipitation is presented rather descriptively. So, it remains unclear, what the connection is between the latitudinal shift of the monsoonal precipitation and an increase in surface winds, in particular, since moist convective cold pools are not resolved in the 60-km model simulations. Here, the authors could strive for a more thorough explanation, possibly, in the context of the West African Monsoon circulation.

   *Answer:*
   We understand this remark. This is our main question for this study. It is difficult to add longer explanation in the abstract (mainly dedicated to summarize the results) but we added a more complete explanation in the conclusion and a new section (with a new Figure) dedicated to the calculation and analysis of the Moist Static Energy:

   *The second set of Figures represents the Moist Static Energy (MSE) at the first model level, which corresponds roughly to a vertical level of 20 m AGL. MSE is defined as:*

   $$MSE = gz + C_p T + Lq \qquad (1)$$

   *with g the gravitational acceleration (g=9.81 m.s$^{-2}$), z the geopotential height (m), $C_p$ the specific heat of dry air at constant pressure (1005 J.kg$^{-1}$.K$^{-1}$), T the temperature (K), L the latent heat of evaporation of water (2256 kJ.kg$^{-1}$) and q the specific humidity (kg.kg$^{-1}$). MSE is a direct indicator of monsoonal precipitation: the transformation of enthalpy and latent heat in the lower troposphere into geopotential energy aloft is a key ingredient of tropical deep convection (Neelin and Held, 1987). The MSE is then expressed in (kJ.kg$^{-1}$) and in the West Africa it is characterised by values around 350-370 kJ.kg$^{-1}$ in the lower troposphere, associated with high temperature and humidity, and around 350 kJ.kg$^{-1}$ at the tropopause, associated with high geopotential, with a minimum in the mid-troposphere (Fontaine and Philippon, 2000). In July 2016, a MSE maximum between 5 and 15ºN well describes the simulated precipitation ( Figure 3 ), while a MSE minimum characterises the dry Sahara.*

   *The MSE response in the CPLanthro simulation shows a significant increase north of 15ºN, associated with the increase in the LW radiation, which accounts for the latent heat flux ( Figure 2 and Figure 3 ). Conversely, the simulated MSE increase south of 15ºN can be explained with the increase in the SW radiation. The MSE response in the CPLdust simulation shows an increase limited to coastal West Africa, while the response in the Sahara region is negative ( Figure 3 ). This pattern can be explained with the simulated LW radiation response shown in Figure 2 . Overall, the differences in the MSE response in the different scenarios account for the simulated differences in the monsoonal precipitation, i.e. a stronger and better organised response of convection in CPLanthro than in CPLdust.*

   The conclusion was changed accordingly.

3. Pages 2, lines 25-34: It might be worth mentioning that also Heinold et al. (2011, doi:10.1111/j.1600-0889.2011.00574.x) had already found very similar effects of Saharan dust and biomass burning smoke strengthening the Hadley circulation, which influenced the aerosol distribution in a similar way as described in this study.

   *Answer:*
   Yes, we completely agree. We missed this article and we are sorry about that, because this study is very close to what we tried to do. In this paper, the focus is more on direct effects, but, that's right, the same conclusions were already quantified in this study. We added this reference in the paper: Heinold et al. (2011).

[Figure]

**Figure 2.** *Left column (from top to bottom): Maps of monthly mean shortwave radiation (down-up, surface) (W/m² ) and longwave radiation (down-up, surface) (W/m²) for July 2016. Middle column: maps of monthly mean of daily averaged differences between CPLanthro and CPLfull for same variables as the left row. Right row: maps of monthly mean of daily averaged differences between CPLdust and CPLfull for same variables as the left row. The tiny crosses on the map indicate where the difference was found to be significant.*

[Figure]

**Figure 3.** *Left column (from top to bottom): Maps of monthly mean rain mixing ratio (kg/kg) and moist static energy (kJ.kg$^{-1}$) for July 2016. Middle column: maps of monthly mean of daily averaged differences between CPLanthro and CPLfull for same variables as the left row. Right row: maps of monthly mean of daily averaged differences between CPLdust and CPLfull for same variables as the left row. The tiny crosses on the map indicate where the difference was found to be significant.*

4. Page 2, line 35: Aerosol particles involved in heterogeneous freezing, today, are more commonly referred to as 'ice nucleating particles (INP)'.

*Answer:*
Ok it was corrected.

5. Pages 5/6, Section 3.2: Does the radiation scheme in the model consider the change in cloud properties due to the aerosol-cloud interactions?

*Answer:*

Yes, absolutely. And the following paragraph was added in the manuscript (section 3.2).

*The change in cloud optical properties due to aerosol-cloud interaction feeds the short and longwave radiations schemes as described in Tuccella et al. (2019). Following Thompson and Eidhammer (2014), the radiative transfer is forced by cloud optical depth calculated from cloud droplet and ice nuclei concentration and their effective radius. The effective radii of liquid and frozen cloud water are computed with the parameterization of Slingo (1989) and Stephens et al. (1990), respectively.*

6. Page 6, Section 4: To my knowledge, extensive aircraft measurements of aerosol chemistry, radiation, and cloud-aerosol interactions took place during DACCIWA. Why were these observations not used for model evaluation?

*Answer:*

Some comparisons to aircraft meaurements were added and are presented above. As already mentioned, we did not use all these aircraft measurements mainly because:

(a) A problem a representativity: the horizontal resolution of the model is too coarse to have a correct comparison with high frequency measurements. Only 2 or 3 horizontal cells of the model cover a whole flight track: we are thus just able to compare a very mean value (with the model) to a very varying value.

(b) The scope of the paper: the article is designed as a model vs model comparison, with scenarions on emissions. The comparison with measurements is made only to see if the model is 'realistic'. It means: is the meteorology correct, are the precipitations well located and, more or less, for the correct period and with the correct intensity. Then, are the surface concentrations and AOD correct? If yes (and here, it is the case), we can be reasonnably confident in the results of the emissions scenarios. This is a model-oriented study such as studies related to regional climate change, for example.

7. Page 21, lines 17-18: The alternating patterns are most likely due to stochastic effects of clouds between the two model representations.

*Answer:*

Yes, it is the correct point of view. This remark was adedd in the text.

8. Fig. 4: The different coloured lines are too thin and hard to distinguish. Since the individual soundings are not discussed anyway in the text, I wonder whether it would make sense to average over the profiles (or groups of them).

*Answer:*

The piece of informations for this Figure is the averaged profiles (bold black line) and the associated standard deviations (grey shade). The different colours are for the different launching sites and we think it is important to distinguish them. The good point with this Reviewer's remark is that the individual profiles are not discussed but should be. Then, a paragraph was added in the text as follows:

*In addition to the mean averaged profiles and their associated standard deviations, the colored profiles correspond to the mean averaged profile for each launching location. It enables to see if systematic biases depends on the launching site location or not. For the wind speed and close to the surface, a positive bias (i.e model overestimates measurements) is diagnosed for Savè and Parakou (Bénin) and not for the other locations, mostly coastal ones, where an underestimation is diagnosed. The boundary layer wind speed is thus too low close to the coast but too large inland.*

and

*the relative humidity is well reproduced in the monsoon layer but there is a dry systematic bias from 1 to 4 km in the simulation, which leads to a probable under-estimation of liquid water content, thus the cloud cover and the indirect effects. This is the case for all soundings, whether the sites are close to the coast or inland.*

Note that the caption of the Figure 4 was also corrected (when re-reading, we found a mistake).

9. Figs. 5 to 7: In the difference plots, the tiny black dots probably indicate statistical significance. This should be mentioned in the figure caption.

   *Answer:*
   OK. The following line was added in the caption of these three Figures: *The tiny crosses on the map indicate where the difference was found to be significant.*

10. Figs. 4 to 8: The font size of axis labels and titles and/or colour bars is too small and needs to be adjusted.

    *Answer:*
    Yes, OK. During the technical check, all Figure font sizes were alread increased, but it seems it is not enough. In this new version, I increased more the font size and I think now it is OK. Note that for the title, it is also provided at the top of each Figure and thus always clearly visible. An example is presented in Figure 4

[Figure]

**Figure 4.** *Exemple of reprocessed Figure with larger ticks and fonts.*

**2  Reviewer #2**

The paper entitled "The role of aerosol-cloud interactions in linking anthropogenic pollution over southern West Africa and dust emission over the Sahara" studies the aerosol direct and indirect effects over West Africa during the DACCIWA in July 2016 using the WRF-CHIMERE coupled model. A reference case is compared against two scenarios with halved emissions of mineral dust and anthropogenic sources, obtaining significant results, even though the impact of the direct and indirect effects is moderate. The paper is a significant contribution to the field and the obtained results are of interest. The paper is well written and the structure is clear. My recommendation is publication after minor revisions.

**General comments:**

The results section is slightly descriptive and a deeper discussion of the results is missing at some points, especially in Section 4 where the modelled data are compared to observations. How does the differences observed here between the model and the data affect the results of the study? What are the uncertainties?

*Answer:*

Complementary informations were added in the Section "4.5 Synthesis of scores". We discuss more each score for each variable and try to project the impact of a bias or an uncertainty on the discussed results. The section now starts with this paragraph:

*The first step is to compare the CPLfull simulation results to available MIDAS, AERONET radiosoundings and aircraft observations. The main goal of this comparison to observations is to assess if the model is realistic enough in order to for us to be confident with the results of the simulations made under different scenarios. The observations are selected because they are spatially homogeneous in the studied area (for MIDAS and AERONET) and automatically acquired so that they are available every day and for the whole studied period. Given the low resolution of the simulation ($\Delta x, y$=60km), we favoured observations with a large spatial extent and for the longer possible period. In addition to these surface observations, we added the vertical soundings acquired during the DACCIWA experiment in Abidjan, Accra, Cotonou, Savè, Lamto and Parakou. Even if they provide a more limited spatial and temporal coverage, they give precious information about the vertical structure of the boundary layer and the troposphere in the studied area. Finally, aircraft measurements are used to assess the quality of some of the key chemical species over southern West Africa ($O_3$, $NO_2$ and $CO$)..*

At the end of this section, we added these concluding remarks:

*These comparisons to observations show that the simulation is realistic for the whole region and during the studied period. The main meteorological spatial strctures are well modelled (for wind speed, temperature and precipitation) even if the day to day variability is not always well represented. The aerosols are mostly well modelled: AOD and ANG scores show that the aerosol mass is realistic with the correct order of magnitude. The aerosol plumes are also spatially well located showing that the conjunction of meteorology and emissions is working well. The simulation is realistic enough to be confident in scenario studies: the differences between the reference case and the scenarios will be spatially well located and with the correct order of magnitude.*

Note that the uncertainty is possible to quantify for an observation (instrumental error, representativeness), but less possible for model results (mean values in cells). Thus, it is possible to quantify errors between model and observations, for a specific domain, resolution, studied period, specific variable, but it is not possible to quantify the uncertainty attached to this result. For modelling, in general, 'uncertainty' is linked to 'variability' and it is obtained by running numerous perturbed simulations (an ensemble).

Additionally, it is necessary to revise the whole manuscript for typos, paying special attention to the references format.

*Answer:*

This problem of references was also pointed out by the Reviewer #1 and the whole manuscript was completely checked. Breifly, we are using two kinds of ways: if we want to describe a work done by Heinold et al. (2011) or if we just want to describe something, (Heinold et al., 2011). The Editorial staff has the source document of this article and this is an editorial's work to adapt the text to their own rules.

**Specific comments:**

1. Page 6, Line 160: Given the importance of biomass burning aerosols, as explained by the authors in the introduction, why is it not included in the analysis?

*Answer:*

Yes, it is a good remark. Indeed, this is accurately presented in the introduction, because in a first version of this paper, we planned to include also a discussion on fires. Finally, this contribution was not included in the paper because:

(a) The results were not a real added value to what is currently discussed.

(b) It added a lot of Figures to the manuscript, making it less understandable.

Finally, the introduction was simplified to be more focussed on anthropogenic and mineral dust emissions impacts. The mention to fires was removed.

2. Page 9, Line 240: Is there any possible explanation for this bias in Savè?

*Answer:*

We added an explanation about this point (assuming the remark refers to line 204). The following senetnce was added in the manuscript: *The order of magnitude is correctly reproduced except in Savè where a low negative bias appears (ranging between 0 and 0.5) associated with fine particles (Ang > 0.5) which suggests an underestimation of anthropogenic aerosol concentration rather than dust or black carbon concentrations*

3. Figure 8: The crosses and wind arrows are difficult to distinguish. Please improve the readability of the figure.

*Answer:*

As for the maps, the vertical cross-section figures were also reprocessed. We increased the size of the symbol "+" and the wind vector. But not too much, because several tries showed that the variables values under the crosses and vectors were hidden.

4. Page 21, line 414: Could you provide a quantitative estimate of this percentage?

*Answer:*

Yes, the conclusion was changed and this sentence is now:

*Two scenarios were used to compute additional simulations with halved emissions of mineral dust and anthropogenic sources. By comparison between the reference case and these scenarios, the direct and indirect effects of aerosol were quantified. Overall, results show a moderate impact of the direct and indirect effects, as also quantified over Europe in Forkel et al. (2015). This impact represents a few percent of the monthly mean value. i.e. $\approx \pm 2\%$ for the boundary layer height, the temperature and wind speed and $\approx \pm 5\%$ for the rain mixing ratio, the shortwave and longwave radiation as well as the maximum of MSE.*

5. Page 21, lines 414-415: "Furthermore, the direct and indirect effects appear to be increasing with time." It is not clear to me how you reach to this conclusion. Please, explain.

*Answer:*

We have no real explanation for this sentence and it was certainly not enough accurate and badly written. It seems that this sentence has no robust foundation. Then, the sentence was removed in this revised manuscript.

**References**

Flamant, C., Knippertz, P., Fink, A. H., Akpo, A., Brooks, B., Chiu, C. J., Coe, H., Danuor, S., Evans, M., Jegede, O., Kalthoff, N., Konaré, A., Liousse, C., Lohou, F., Mari, C., Schlager, H., Schwarzenboeck, A., Adler, B., Amekudzi, L., Aryee, J., Ayoola, M., Batenburg, A. M., Bessardon, G., Borrmann, S., Brito, J., Bower, K., Burnet, F., Catoire, V., Colomb, A., Denjean, C., Fosu-Amankwah, K., Hill, P. G., Lee, J., Lothon, M., Maranan, M., Marsham, J., Meynadier, R., Ngamini, J.-B., Rosenberg, P., Sauer, D., Smith, V., Stratmann, G., Taylor, J. W., Voigt, C., and Yoboué, V.: The Dynamics-Aerosol-Chemistry-Cloud Interactions in West Africa Field Campaign: Overview and Research Highlights, Bulletin of the American Meteorological Society, 99, 83–104, https://doi.org/10.1175/BAMS-D-16-0256.1, 2018.

Fontaine, B. and Philippon, N.: Seasonal evolution of boundary layer heat content in the West African monsoon from the NCEP/NCAR reanalysis (1968-1998), International Journal of Climatology, 20, 1777–1790, https://doi.org/10.1002/1097-0088(20001130)20:14<1777::AID-JOC568>3.0.CO;2-S, 2000.

Forkel, R., Balzarini, A., Baro, R., Bianconi, R., Curci, G., Jimenez-Guerrero, P., Hirtl, M., Honzak, L., Lorenz, C., Im, U., Perez, J. L., Pirovano, G., Jose, R. S., Tuccella, P., Werhahn, J., and Zahbkar, R.: Analysis of the WRF-Chem contributions to AQMEII phase2 with respect to aerosol radiative feedbacks on meteorology and pollutant distributions, Atmospheric Environment, 115, 630 – 645, https://doi.org/https://doi.org/10.1016/j.atmosenv.2014.10.056, 2015.

Heinold, B., Tegen, I., Bauer, S., and Wendisch, M.: Regional modelling of Saharan dust and biomass-burning smoke, Tellus B, 63, 800–813, https://doi.org/10.1111/j.1600-0889.2011.00574.x, 2011.

Knippertz, P., Fink, A. H., Deroubaix, A., Morris, E., Tocquer, F., Evans, M. J., Flamant, C., Gaetani, M., Lavaysse, C., Mari, C., Marsham, J. H., Meynadier, R., Affo-Dogo, A., Bahaga, T., Brosse, F., Deetz, K., Guebsi, R., Latifou, I., Maranan, M., Rosenberg, P. D., and Schlueter, A.: A meteorological and chemical overview of the DACCIWA field campaign in West Africa in June–July 2016, Atmospheric Chemistry and Physics, 17, 10 893–10 918, https://doi.org/10.5194/acp-17-10893-2017, 2017.

Neelin, J. D. and Held, I. M.: Modeling Tropical Convergence Based on the Moist Static Energy Budget, Monthly Weather Review, 115, 3–12, https://doi.org/10.1175/1520-0493(1987)115<0003:MTCBOT>2.0.CO;2, 1987.

Slingo, A.: A GCM Parameterization for the Shortwave Radiative Properties of Water Clouds, Journal of the Atmospheric Sciences, 46, 1419–1427, https://doi.org/10.1175/1520-0469(1989)046<1419:AGPFTS>2.0.CO;2, 1989.

Stephens, G. L., Tsay, S.-C., Stackhouse, P. W., and Flatau, P. J.: The Relevance of the Microphysical and Radiative Properties of Cirrus Clouds to Climate and Climatic Feedback, Journal of the Atmospheric Sciences, 47, 1742–1754, https://doi.org/10.1175/1520-0469(1990)047<1742:TROTMA>2.0.CO;2, 1990.

Sultan, B. and Janicot, S.: The West African Monsoon Dynamics. Part II: The Preonset and Onset of the Summer Monsoon, Journal of Climate, 16, 3407–3427, https://doi.org/10.1175/1520-0442(2003)016<3407:TWAMDP>2.0.CO;2, 2003.

Thompson, G. and Eidhammer, T.: A study of aerosols impacts on clouds and precipitation development in a large winter cyclone, J Atmos Sci, 71, 3636–3659, https://doi.org/10.1175/JAS-D-13-0305.1, 2014.

Tuccella, P., Menut, L., Briant, R., Deroubaix, A., Khvorostyanov, D., Mailler, S., Siour, G., and Turquety, S.: Implementation of Aerosol-Cloud Interaction within WRF-CHIMERE Online Coupled Model: Evaluation and Investigation of the Indirect Radiative Effect from Anthropogenic Emission Reduction on the Benelux Union, JOURNAL = Atmosphere, 10, https://doi.org/10.3390/atmos10010020, 2019.